**Data Availability Statement:** All relevant data are within the paper and its Supporting information file.

**Funding:** The author(s) received no specific funding for this work.

# Large private shareholders, industrial policies and industrial loans of city commercial banks: Evidence from China

**Jie Liu**◉*, **Qiaoyun Zhang, Chao Xiao**

School of Finance, Southwestern University of Finance and Economics, Chengdu, Sichuan, China

* liuj@smail.swufe.edu.cn

## Abstract

We show that large private shareholders have an information advantage about their industry; this can alleviate the information asymmetry suffered by banks, and consequently, increase bank lending to these shareholders' industry. Using a sample of Chinese city commercial banks, we show that an increase in the large private shareholders' shareholding of banks increases bank lending to these shareholders' industry. Importantly, using Chinese local government industrial policy as a moderator, we find that industrial policies have a positive and significant moderating effect on the relationship between large private shareholders and banks' industry-specific lending. This relationship strengthens when local industrial policy supports these shareholders' industry. This helps explain why banks prefer the industries to which their large private shareholders belong to and how industrial policy affects bank credit allocation.

## Introduction

The influence of large shareholders on bank loans is mixed. Large shareholders may be more active [1, 2], but they may also be more cautious [3]. In addition, the cyclical nature of bank loans is relevant to large shareholders. Large state-owned shareholders are often counter cyclical, while large private shareholders are more pro-cyclical [4–6]. However, Berrospide and Edge found that the change of bank capital has little impact on bank loans, which to a certain extent indicates that the change of shareholders is not the key to affecting bank loans [7].

These mixed findings mostly study the impact of large state-owned shareholders on the total scale of bank loans. However, the state-owned shareholders are not independent (directly intervened by the government) [8–11], which leads to the phenomenon that the bank lending behavior caters to the government preference. This cannot reflect the basic impact of other characteristics of large shareholders (industry background and shareholding) on bank loans. Additionally, the influence of the industry attribute of major shareholders on bank loans should be reflected in the preference of banks for a certain industry, which unfortunately has received little attention.

**Competing interests:** The authors have declared
that no competing interests exist.

Generally, large shareholders have a greater impact than other shareholders. Large private shareholders are more market-oriented, which ensures that banks are subject to less direct government intervention [11]. Then, can large private shareholders give banks special information about their industry? Does this information advantage lead banks to lend more to this industry? Is the impact of large private shareholders on banks' industry-specific loans affected by macro-level policies? These questions can help explain why shareholders can have delegated monitoring and informational advantages in financial intermediaries, and what drives banks to lend more to one industry instead of another.

We also focus on testing the moderating effect of industrial policies on shareholders' influence on bank lending. The government uses industrial policies to guide industrial investment, optimize the economic structure, and promote industrial upgrading [12]. Industrial policy is embodied in the government's encouragement, guidance, or restriction of capital flows to specific industries [13, 14]. For instance, positive industry information can be provided to the market by publishing the supported industries' information [15]. This provides the signal of a government "guarantee" to a certain extent [16]. Meanwhile, guiding and restricting the flow of funds affect the industrial distribution of bank loans to industries which are affected by the industrial policies. However, this effect is not direct. Shareholders ultimately determine bank lending. Thus, we use industrial policy as a moderator and test its effect on the impact of large private shareholders on banks' industrial loans.

Chinese city commercial banks are an interesting research setting because of China's banking reforms. To diversify the shareholding structure, China has repeatedly encouraged mixed ownership reforms of banking institutions and continuously expanded the channels for private enterprises to be shareholders in banks. City commercial banks are the most important type of bank serving private enterprises in China, and intuitively, closely related to these enterprises. Their scale is much smaller than that of large state-owned banks. Thus, private enterprises with lower financial strengths prefer to hold shares of city commercial banks. We analyzed the data of city commercial banks from 2007 to 2020 and found that the number of private enterprise shareholders is significantly higher than that of state-owned enterprise shareholders among the top ten shareholders (excluding local governments, such as finance bureaus and state-owned asset management companies). This significantly differs the large state-owned banks, which have few private shareholders. In addition, the moderating influence of local industrial policy is usually local. Similarly, city commercial banks mainly help local economic development. Therefore, using city commercial banks as our sample ensures that we accurately test the influence of large private shareholders on banks' industry-specific loans and the moderating effect of industrial policies.

We empirically test this relationship considering industry attributes and the shareholding of large private shareholders of city commercial banks, and the industries that are supported or explicitly encouraged by the local "Five-Year Plan". We find that industry attributes and shareholding of large private shareholders significant impact the industry-specific loans of city commercial banks. Specifically, as the shareholding of large private shareholders increases, city commercial banks make more loans to industries to which these shareholders belong to. Importantly, local governments' industrial policies significantly moderates the impact of large private shareholders: the impact is stronger when industrial policies support these shareholders' industry.

There are four main contributions of our study:

First, there has been a wealth of research on the impact of ownership structure on loan behavior. [2, 4, 5, 7], we innovatively study the impact of large private shareholders on the allocation of credit resources in city commercial banks. Studies have examined bank credit behavior from the perspective of total loan volume or term structure [9, 17, 18], and less from the

perspective of industry-wise distribution of bank loans. However, this distribution of loans can be a more comprehensive demonstration of how banks allocate their credit resources. We provide a new perspective for examining bank lending by studying the impact of large private shareholders on banks' industry-specific lending behavior.

Second, we indirectly study the impact of industrial policies on bank loans. Bank lending behavior is ultimately determined by the board of directors and management, which in turn depends on the bank's shareholding structure. Therefore, we define industrial policy as a moderating factor and focus on analyzing its moderating effect on the impact of large private shareholders on bank lending to these shareholders' industry. Our discussion enriches the research on the economic consequences of industrial policy [13, 14, 16].

Third, studies on the allocation of credit resources by industrial policy are based more on the availability of credit to enterprises [15, 19]. Our research, which is based on banks' industry loans, can more accurately reflect the industrial allocation of credit resources affected by industrial policies. Therefore, we elucidate one mechanism through which industrial policies affect banks' industrial loans.

Fourth, there have been a large number of literature studies on the impact of state-owned shareholders [4–6], in contrast, another outstanding contribution of ours is to analyze large private shareholders. This is helpful to understand the industry expertise of private shareholders and their role in bank lending decisions. This also enriches the theory of information asymmetry.

## Literature review, theoretical analysis and research hypothesis

### Ownership expropriation and industrial loan in Chinese city commercial banks

China's banking industry often has government holdings and suffers from government intervention. This negatively affects bank operations and brings inefficiencies into bank credit allocation [8, 20–22]. However, the ability of local governments to intervene in the local branches of state-owned banks gradually has declined as the branches of state-owned banks have broken away from their affiliation with local governments (e.g., the headquarters of large state-owned banks abolished the branches' credit allocation authority). Nevertheless, local governments still need to undertake regional infrastructure construction and public expenditure; thus, city commercial banks have become an important source of funds for local governments to support local economic development. However, the local government's intervention distorts efficient, market-based operations and credit allocation of city commercial banks; this increases banks' non-performing loans and they have a low ability to resist risks [23, 24]. This is due to the differences in mandates of banks and the government: government shareholders are more concerned with providing employment, investment in public goods, social welfare, among others. This "political view" means that banks' credit allocation is not based on economic benefits. Ultimately, this increases bank risks, and restricts financial development and economic growth [25–28].

The influence of state-owned enterprise shareholders is similar to that of government shareholders because the former are controlled by the government. The inefficient allocation of credit resources caused by state-owned shareholders accelerates the depletion of bank capital. This forces banks to reduce lending and allocate to risky assets [7, 17, 18]. To enhance the ability of city commercial banks to serve local economic development, various regions have stepped up efforts to introduce strategic investors. Many city commercial banks have welcomed private shareholders as strategic investors. Different types of shareholders can help realize the diversified ownership structure reforms, and form a check and balance ownership

**Table 1. Group differences in the shareholding of large private shareholders.**

| | Before the introduction of regulatory policy | | After the introduction of regulatory policy | | | |
|---|---|---|---|---|---|---|
| | N | Mean | N | Mean | Mean Difference | t Value |
| *LPS* | 335 | 0.069 | 881 | 0.082 | -0.014*** | -4.112 |

This table reports the group differences in the shareholding of large private shareholders of city commercial banks before and after the Policy Document *No*.27. Variables are defined in Table A1 in S1 Appendix.

***, **, * indicate statistical significance at the 1%, 5%, and 10% levels, respectively.

structure. This can optimize the level of corporate governance and enhance banks' ability to resist risks.

In terms of regulations, in 2012, the former CBRC issued Policy Document *No*.27 to support outstanding private enterprises to invest in banking and financial institutions. City commercial banks were one of the main targets. Table 1 reports reports the shareholding of large private shareholders before and after the introduction of the regulatory policy. The mean shareholding of large private shareholders increased from 6.9% to 8.2%, with the difference being significant at the 1% level. Thus, the policy seems to have been effective.

In fact, there is a big difference between the industries of large private and state-owned shareholders (as shown in Table A2 in S1 Appendix). The influence of large private shareholders on banks' industry loans is less researched. However, this topic is interesting.

## Theoretical analysis

The "Market for Lemons" model proposed by Akerlof [29] pioneered the theory of information asymmetry. Information asymmetry embodies the idea that "sellers are better than buyers". For example, in credit markets, there are differences in the information available to banks and borrowers. Borrowers (banks) with better (a lack of) information have an advantage (a disadvantage) in transactions. Various problems and risks caused by information asymmetry are particularly prominent and serious in developing countries which are transitioning to market economies.

The common consequences of information asymmetry are adverse selection (hidden information) and moral hazard (hidden action). Adverse selection was first formally analyzed by Mirrlees [30]. In terms of credit behavior, high-risk borrowers tend to look more for loans because they will reap huge returns if their investment projects are successful. However, these borrowers are also the most likely to default. When banks do not understand the risks imposed by their borrowers, they may not lend correctly; that is, they may lend to these high-risk borrowers without appropriately hedging against/accounting for the risks. Meanwhile, moral hazards occur after a loan is issued. Banks cannot effectively supervise borrowers' behavior because banks and borrowers are only in debt-creditor relationships. Borrowers may harm banks for their own benefit [31]. For example, borrowers may use loans for high-risk investments that banks do not support because these investments are more likely to fail; this can increase the risk of loan defaults. Then, banks may reduce lending because they realize that such risks can lead to loan default.

The same is true for an industry. A bank's knowledge of a certain industry determines whether companies in the same industry can obtain loans from the bank [32]. This ultimately affects whether banks prefer lending to that industry. If the existence of large private shareholders can familiarize the bank with information on these shareholders' industry, then the bank may issue more loans to that industry.

## Hypotheses

Information about borrowers is fundamental for bank lending [33]. As noted, banks may suffer serious adverse selection and moral hazard issues due to information asymmetry [31, 34]; therefore, they may reduce lending to particular borrowers. Generally, banks can only obtain the basic financial information of borrowers; it is difficult to obtain private information, such as corporate structure and human capital [35]. Large private shareholders possess industry-specific information, can accurately identify the development status of their industry, and keep abreast of the details of related companies. Then, the existence of large private shareholders enables banks to more effectively identify borrowers' financial information in that industry and have a certain understanding of special information. These shareholders can then improve the bank's ability to effectively identify which companies in their industry are "bad" borrowers, thereby mitigating adverse selection.

Large private shareholders also play an indispensable role after loans are issued. Traditionally, banks hold borrower ownership [36, 37], require security and collateral [38–40] or urge borrowers to disclose information [41] to curb moral hazard. Meanwhile, large private shareholders play a different role in restraining moral hazards. On the one hand, banks can leverage the information advantages of large private shareholders to formulate debt contracts that can constrain borrowers' behavior better after they obtain loans. On the other hand, large private shareholders may continue to have more sensitive and accurate judgments on their industry. This allows banks to dynamically and effectively supervise borrowers' behavior, which helps mitigate moral hazard.

Thus, compared with other industries, banks have an information advantage about large private shareholders' industries. Because the information held by these shareholders on their industry is often proprietary, it can yield insights into the industry that are not available from other sources. These insights can reduce banks' information costs for companies with similar industry backgrounds and improve lending to these companies.

Accordingly, we propose the Hypothesis 1 as follows:

**Hypothesis 1**. The relationship between the shareholding of large private shareholders and bank loans to their industries is positive.

Industrial policies guide the flow of bank loans, realize the allocation of funds to key industries, and thereby promote the adjustment of the industrial structure. Enterprises that are selected and supported by industrial policies generally have high growth prospects. Thus, information that is otherwise difficult to obtain (such as enterprise growth capabilities) is transmitted to banks, which alleviates the information asymmetry between banks and enterprises [13, 14]. Further, government guidance and intervention can change the development prospects and investment behavior of enterprises. For instance, it can improve the total factor productivity of enterprises and the entire industry, which further enhances bank loan support for these industries [19, 42, 43]. In addition, companies can use the industrial policy support as a signal to banks, which is actually an invisible guarantee from the government. Furthermore, companies that receive policy support are usually encouraged and supervised by the government, which makes it easier for them to obtain bank loans [15, 16].

Of course, this does not represent that industrial policies can significantly promote bank loans to related industries. The bank's lending behavior is determined by the board of directors, which ultimately depends on the bank's shareholding structure.

Accordingly, we propose the Hypothesis 2 as follows:

**Hypothesis 2**. There is no direct correlation between industrial policies and bank loans.

However, industrial policy can promote the impact of large private shareholders on banks' industrial loans. These shareholders can provide an information advantage if their industries are also supported by the industrial policy; thus, banks increasingly believe that these industries are credible. In other words, industrial policies have moderating effects. This strengthens the influence of large private shareholders, and more bank loans are likely to be issued to these industries.

Accordingly, we propose the Hypothesis 3 as follows:

**Hypothesis 3**. The moderating effect of industrial policies would strengthen the positive relationship between the shareholding of large private shareholders and bank loans to their industries.

## Data and methods

### Data

We select 126 Chinese city commercial banks, and the sample is the unbalanced panel data from 2007 to 2020. The shareholding structure, the industry loan and other financial data of the sample banks are partly sourced from the Wind database, and the data of unlisted banks are collected and sorted manually according to the annual reports published on the official website of the banks. The macro data to measure regional economic characteristics come from the Wind database and the National Bureau of Statistics of China.

The industrial policies data mainly comes from the "Outline of the Eleventh Five-Year Plan for National Economic and Social Development" (2006–2010), "Outline of the Twelfth Five-Year Plan for National Economic and Social Development" (2011–2015) and "Outline of the Thirteenth Five-Year Plan for National Economic and Social Development" (2016–2020) issued by provincial governments in China.

### Model specification

In order to empirically test the impact of large private shareholders on banks' industrial loans and the moderating effect of industrial policies, the OLS regression equation with fixed effects is as follows.

$$IndLoan_{it} = \beta_0 + \beta_1 LPS_{it} + \beta_2 IP_{it} + \beta_3 IP_{it} \times LPS_{it} + \theta X + \omega_{it} + \eta_t + \epsilon_{it}, \tag{1}$$

### Measures

The dependent variable in Eq (1), $IndLoan_{it}$, is the ratio of bank $i$'s loans to the industry of large private shareholder to total loans in year $t$. The independent variable, $LPS_{it}$, is the shareholding of large private shareholder of bank $i$ in year $t$. The moderating variable is a dummy, $IP_{it}$. If the industries that are supported by the industrial policies include large private shareholders, the $IP$ is equal to 1, otherwise it is 0. $\beta_0$ is a constant term, $\omega_{it}$ is the industry fixed effect of the large private shareholders, $\eta_t$ is the year fixed effect, and $\epsilon_{it}$ is the error term. $X$ are some control variables, including bank characteristics and macroeconomic characteristics.

Characteristic variables at the bank level: *Size*, *LDR*, *Fore*, *CAR* and *Sta*. Characteristic variables of regional macroeconomics: *GDPr*, *Deptr*, *SOE*, *GDP_sec* and *GDP_tert*. Variables are defined in Table A1 in S1 Appendix.

## Results and discussion

### Descriptive statistics

Table 2 reports descriptive statistics. The mean of *IndLoan* is 0.16. That is banks' loans to the industry of the large private shareholder accounted for 16% of the total loans on average. The mean of *LPS* is 0.08, indicating that the large private shareholder owns 8% of the bank's ownership stakes on average. The mean of *IP* is 0.62, indicating that 62% of the large private shareholders of the sample banks belong to industries supported by industrial policies.

Table 3 reports the correlation matrix. The correlation coefficient between *LPS* and *IndLoan* was 0.247 and was highly significant. It shows that the increase in the shareholding of large private shareholders has increased bank lending to their industries. This preliminarily confirmed our **Hypothesis 1**.

We plot the relationship between the shareholding of large private shareholders and the bank's industry loans, and the results are shown in Fig 1. The line is inclined to the upper right, indicating that this relationship is positively correlated. Fig 1 can preliminarily confirm that large private shareholders can influence the lending behavior of city commercial banks.

We can preliminarily judge the moderating effect of industrial policies as shown in Fig 2. The case of large private shareholders who are supported by industrial policy is a solid line, and the case that is not supported by industrial policy is a dotted line. It can be found that both the solid line and the dashed line are inclined to the upper right. It shows that in both cases, large private shareholders can affect the bank's industry loans, which is consistent with Fig 1. However, the large private shareholders supported by industrial policies have more influence.

Figs 1 and 2 confirm our **Hypothesis 1** and **Hypothesis 3** to a certain extent, and then we further test them through rigorous empirical analysis.

### Results

Combining Eq (1), we use an ordinary least squares (OLS) regression to regress bank loans on large private shareholders' industry and shareholding, and test the moderating effect of industrial policies. Table 4 reports these results. Column 1 shows that the impact of large private shareholders is positive and significant. That is, higher the shareholding of large private shareholders, the

**Table 2. Descriptive statistics.**

| Variables | N | Mean | Sd | Min | P50 | Max |
|---|---|---|---|---|---|---|
| *IndLoan* | 1216 | 0.16 | 0.13 | 0.00 | 0.12 | 0.78 |
| *LPS* | 1216 | 0.08 | 0.05 | 0.00 | 0.07 | 0.50 |
| *IP* | 1216 | 0.62 | 0.49 | 0.00 | 1.00 | 1.00 |
| *Size* | 1216 | 25.43 | 1.09 | 22.40 | 25.39 | 28.70 |
| *LDR* | 1216 | 0.63 | 0.13 | 0.17 | 0.64 | 1.58 |
| *Fore* | 1216 | 0.29 | 0.46 | 0.00 | 0.00 | 1.00 |
| *CAR* | 1216 | 0.13 | 0.04 | 0.06 | 0.13 | 0.60 |
| *Sta* | 1216 | 0.04 | 0.21 | 0.00 | 0.00 | 1.00 |
| *GDPr* | 1216 | 0.08 | 0.03 | -0.03 | 0.08 | 0.17 |
| *Deptr* | 1216 | 0.13 | 0.06 | -0.05 | 0.12 | 0.40 |
| *SOE* | 1216 | 0.32 | 0.32 | 0.05 | 0.26 | 2.91 |
| *GDP_sec* | 1216 | 0.46 | 0.07 | 0.16 | 0.46 | 0.61 |
| *GDP_tert* | 1216 | 0.45 | 0.08 | 0.29 | 0.45 | 0.83 |

This table reports the summary statistics. Variables are defined in Table A1 in S1 Appendix. The sample period is from 2007 to 2020.

**Table 3. Correlation matrix.**

| | 1 | 2 | 3 | 4 | 5 | 6 | 7 |
|---|---|---|---|---|---|---|---|
| 1 IndLoan | 1 | | | | | | |
| 2 LPS | 0.247*** | 1 | | | | | |
| 3 IP | 0.345*** | 0.134*** | 1 | | | | |
| 4 Size | -0.218*** | -0.266*** | -0.034 | 1 | | | |
| 5 LDR | 0.080*** | -0.025 | 0.044 | 0.189*** | 1 | | |
| 6 Fore | -0.054* | -0.122*** | -0.059** | 0.353*** | 0.109*** | 1 | |
| 7 CAR | -0.001 | 0.021 | 0.047* | -0.211*** | -0.016 | -0.056* | 1 |
| 8 Sta | -0.087*** | -0.124*** | 0.028 | 0.057** | -0.049* | 0.036 | 0.069** |
| 9 GDPr | -0.017 | -0.134*** | 0.022 | -0.349*** | -0.227*** | -0.008 | 0.081*** |
| 10 Deptr | 0.008 | -0.127*** | 0.043 | -0.287*** | -0.207*** | 0.005 | 0.025 |
| 11 SOE | -0.134*** | -0.072** | -0.039 | 0.267*** | 0.151*** | 0.035 | -0.001 |
| 12 GDP_sec | 0.126*** | 0.017 | 0.042 | -0.459*** | -0.217*** | -0.053* | 0.028 |
| 13 GDP_tert | -0.041 | -0.014 | 0.049* | 0.530*** | 0.348*** | 0.168*** | -0.063** |
| | 8 | 9 | 10 | 11 | 12 | 13 | |
| 1 IndLoan | | | | | | | |
| 2 LPS | | | | | | | |
| 3 IP | | | | | | | |
| 4 Size | | | | | | | |
| 5 LDR | | | | | | | |
| 6 Fore | | | | | | | |
| 7 CAR | | | | | | | |
| 8 Sta | 1 | | | | | | |
| 9 GDPr | 0.087*** | 1 | | | | | |
| 10 Deptr | -0.011 | 0.577*** | 1 | | | | |
| 11 SOE | 0.070** | -0.292*** | -0.115*** | 1 | | | |
| 12 GDP_sec | -0.095*** | 0.589*** | 0.467*** | -0.361*** | 1 | | |
| 13 GDP_tert | 0.028 | -0.606*** | -0.466*** | 0.351*** | -0.862*** | 1 | |

This table reports the correlation matrix. Variables are defined in Table A1 in S1 Appendix. The sample period is from 2007 to 2020.

more banks lend to these shareholders' industry. This is similar to the study of Liu et al. [44]. In Column 2, the effect of *IP* is positive, but not significant. We test the moderating effect of industrial policies by interacting *IP* with *LPS*. We find a positive and significant coefficient. This shows that industrial policy is a significant moderator that can enhance the influence of large private shareholders on a bank's industrial loans. Thus, **Hypothesis 1–3** are supported.

We pursue a more visual moderating effect of industrial policies, therefore, we refer to the method of Dawson [45] and draw a moderation plot based on Table 4 as shown in Fig 3. It shows that the relationship between large private shareholders and banks' loans to their industries is always positive, but is more important for industries supported by industrial policy (the dotted line) than those not supported (the solid line).

## Robustness check

We consider a host of robustness tests to corroborate the moderating effect of local industrial policies.

**City commercial banks operate across provinces.** Some city commercial banks have established branches across provinces. Hence, the operation of these banks is also affected by

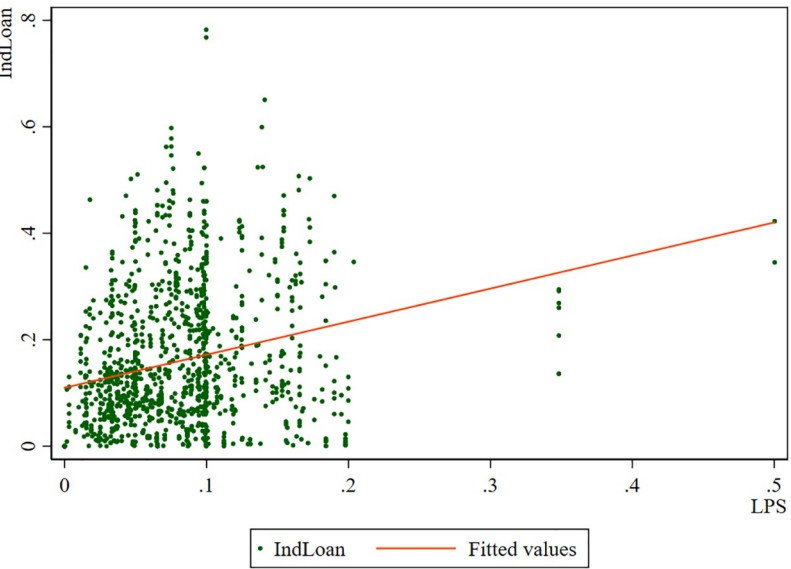

**Fig 1. The shareholding of large private shareholders and banks' industry loans.**

the industrial policies of the locations where the branches are located; moreover, industrial policies may differ across regions. However, industrial policies in the region where the head office of the bank is located cannot affect the loans of branches in other regions.

We group the sample according to whether city commercial banks operate across provinces and conduct a robustness test on the moderating effect of industrial policies. Table 5 reports these results. In column 1, the sample is banks with branches across provinces. The coefficients of the interaction term of *IP* and *LPS* are not significant. In column 2, the sample is banks that do not have branches across provinces. The coefficient of the interaction term is positive and significant. The results show that the industrial policy of the region where the head office is

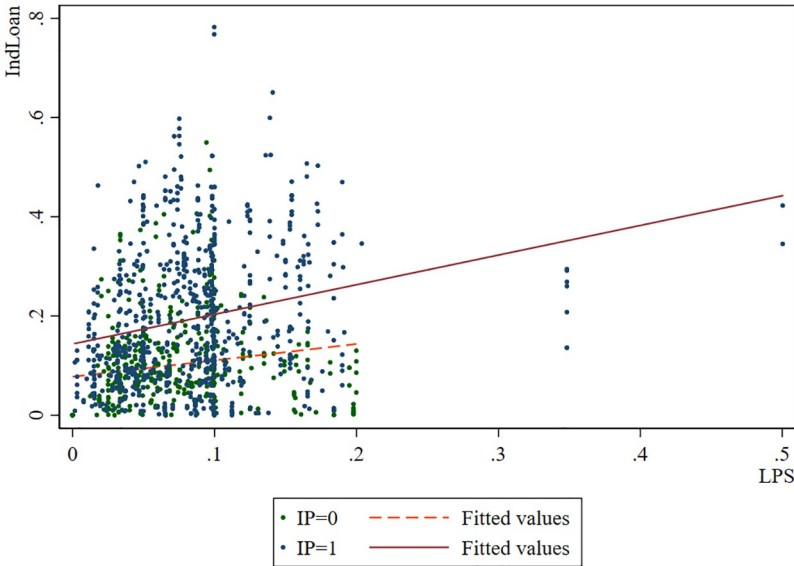

**Fig 2. The shareholding of large private shareholders, local industrial policies and banks' industry loans.**

**Table 4. Large private shareholders, industrial policies and industrial loans of city commercial banks.**

| | (1) | (2) |
|---|---|---|
| | *IndLoan* | *IndLoan* |
| IP × LPS | | 0.420*** |
| | | (3.18) |
| LPS | 0.288*** | 0.228*** |
| | (4.69) | (3.55) |
| IP | | 0.00355 |
| | | (0.46) |
| Size | -0.0160*** | -0.0169*** |
| | (-4.23) | (-4.46) |
| LDR | 0.153*** | 0.148*** |
| | (6.05) | (5.83) |
| Fore | 0.00639 | 0.00456 |
| | (0.91) | (0.65) |
| CAR | -0.105 | -0.0903 |
| | (-1.28) | (-1.10) |
| Sta | 0.00578 | 0.00395 |
| | (0.38) | (0.26) |
| GDPr | -0.567*** | -0.571*** |
| | (-3.08) | (-3.11) |
| Deptr | -0.00275 | 0.00261 |
| | (-0.03) | (0.03) |
| SOE | -0.0223* | -0.0232* |
| | (-1.67) | (-1.74) |
| GDP_sec | 0.273*** | 0.292*** |
| | (2.98) | (3.19) |
| GDP_tert | 0.206** | 0.233*** |
| | (2.36) | (2.66) |
| _cons | 0.201* | 0.207* |
| | (1.71) | (1.75) |
| industry/year | Yes | Yes |
| N | 1216 | 1216 |
| F | 29.47 | 28.38 |
| r2 | 0.474 | 0.478 |

This table reports the results of OLS regressions analyzing the impact of large private shareholders on industrial loans of city commercial banks and the moderating effect of industrial policies. Columns 1 and 2 use *IndLoan* as the dependent variable. Variables are defined in Table A1 in S1 Appendix. The sample period is from 2007 to 2020. In parentheses are t-statistics.

***, **, * indicate statistical significance at the 1%, 5%, and 10% level, respectively.

located does not have a moderating effect for city commercial banks with branches across provinces. This is because different industrial policies in other regions inhibit the moderating effect of local industrial policies.

**Consider the timeliness of industrial policies.** We exclude the effect of timeliness because it takes time for banks to respond to industrial policies. We exclude the sample from the years in which industrial policies were introduced (2011 and 2016). Further, we adopt a lag period for the *IP*. Table 6 lists the results of the robustness tests. In column 1, we delete the

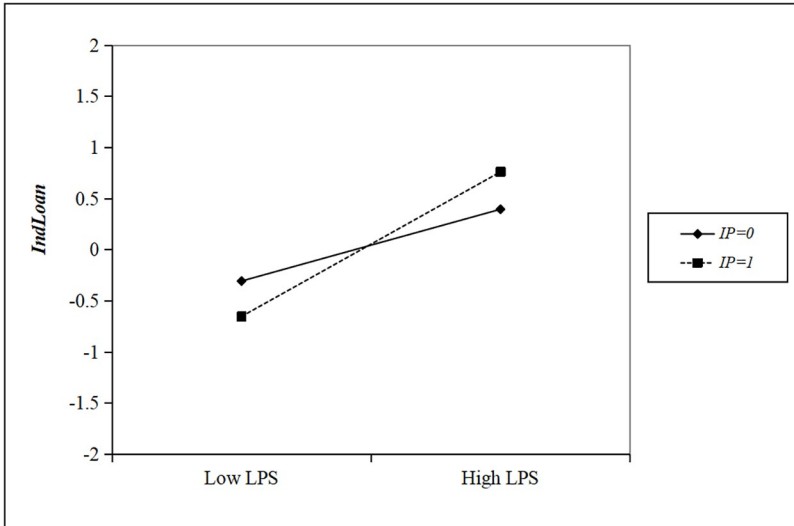

**Fig 3. Moderating effect of local industrial policies on the large private shareholders—Industry loans of city commercial banks relationship.**

sample of the years in which the policies were introduced. The coefficient of the interaction term of *IP* and *LPS* is positive and significant. In column 2, the moderator is the lag term of *IP*, and the coefficient of the interaction term is still positive and significant.

**Consider the turnover of local officials.** The implementation of industrial policies may change before or after the turnover of local officials because these officials play an important role in influencing banks' lending behavior in China. We manually collate data on the turnover of local officials (party secretaries and governors of provinces, municipalities directly

**Table 5. Large private shareholders, local industrial policies and industrial loans of city commercial banks (consider cross-provincial operations of banks).**

|  | (1) | (2) |
|---|---|---|
|  | **Banks that have established branches across provinces *IndLoan*** | **Banks that have not established branches across provinces *IndLoan*** |
| *IP × LPS* | 0.527 | 0.494*** |
|  | (1.24) | (3.36) |
| *LPS* | 0.124 | 0.244*** |
|  | (0.62) | (3.44) |
| *IP* | 0.0261 | -0.00552 |
|  | (1.45) | (-0.59) |
| control variables/ industry/year | Yes | Yes |
| *N* | 278 | 938 |
| F | 10.83 | 20.37 |
| r2 | 0.610 | 0.463 |

This table reports the results of OLS regression analyzing the moderating effect of local industrial policies considering the cross-provincial operation of banks. Columns 1 and 2 use *IndLoan* as the dependent variable. Variables are defined in Table A1 in S1 Appendix. The sample period is from 2007 to 2020. In parentheses are t-statistics.

***, **, * indicate statistical significance at the 1%, 5%, and 10% level, respectively.

**Table 6. Large private shareholders, local industrial policies and industrial loans of city commercial banks (consider the timeliness of industrial policies).**

|  | (1) | (2) |
|---|---|---|
|  | *IndLoan* | *IndLoan* |
| $IP \times LPS$ | 0.408*** |  |
|  | (2.79) |  |
| $IP_{t-1} \times LPS$ |  | 0.354*** |
|  |  | (2.75) |
| LPS | 0.218*** | 0.257*** |
|  | (3.09) | (3.96) |
| IP | 0.00392 |  |
|  | (0.46) |  |
| $IP_{t-1}$ |  | -0.00551 |
|  |  | (-0.74) |
| control variables/industry/year | Yes | Yes |
| N | 1021 | 1076 |
| F | 24.43 | 27.47 |
| r2 | 0.472 | 0.488 |

This table reports the results of OLS regression analyzing the moderating effect of local industrial policies considering the timeliness of industrial policies. Columns 1 and 2 use *IndLoan* as the dependent variable. Variables are defined in Table A1 in S1 Appendix. The sample period is from 2007 to 2020. In parentheses are t-statistics. ***, **, * indicate statistical significance at the 1%, 5%, and 10% level, respectively.

under the central government, and autonomous regions) to test the robustness of the moderating effects.

We control for the influence of local officials in Table 7. In column 1, we control for the effect of party secretaries' turnover. In column 2, we control for the effect of governors' turnover. In column 3, we control for the effect of both party secretaries and governors. The coefficients of the interaction terms are all positive and significant.

**Difference between groups.** DID estimation method is commonly used in policy evaluation. Unfortunately, industrial policy changes in time dimension, it is difficult to estimate them through classic DID. Therefore, we group the sample according to the criteria of whether the large private shareholders are supported by industrial policies. The impact of large private shareholders on banks' industrial loans is tested in two groups. Table 8 reports these results. The large private shareholders of the banks in the sample in column 1 are not supported by industrial policies (*IP*=0). The coefficient of *LPS* is 0.145, which is significant. The large private shareholders of the banks in the sample in column 2 are supported by industrial policies (*IP*=1). The coefficient of *LPS* is 0.518 and is highly significant. Difference between groups is tested by Seemingly Unrelated Estimation. The Chi2 is 6.78 and the P-value is 0.0092. This indicates that there is a significant difference between groups in the coefficient of *LPS*. It shows that the large private shareholders supported by the industrial policies have a stronger impact on the bank's industrial loans (0.518>0.145).

## Conclusion

We argue that large private shareholders can alleviate the information disadvantage that city commercial banks suffer in the credit market by providing proprietary industry information; therefore, banks increase lending to these shareholders' industries. We focus on Chinese city

**Table 7. Large private shareholders, local industrial policies and industrial loans of city commercial banks (consider the turnover of local officials).**

| | (1) | (2) | (3) |
|---|---|---|---|
| | *IndLoan* | *IndLoan* | *IndLoan* |
| $IP \times LPS$ | 0.419*** | 0.420*** | 0.419*** |
| | (3.17) | (3.18) | (3.17) |
| *LPS* | 0.228*** | 0.227*** | 0.229*** |
| | (3.56) | (3.54) | (3.56) |
| *IP* | 0.00329 | 0.00355 | 0.00328 |
| | (0.43) | (0.46) | (0.43) |
| turnover of the party secretary | Yes | No | Yes |
| turnover of the governor | No | Yes | Yes |
| control variables/industry/year | Yes | Yes | Yes |
| *N* | 1216 | 1216 | 1216 |
| F | 27.69 | 27.63 | 26.98 |
| r2 | 0.479 | 0.478 | 0.479 |

This table reports the results of OLS regression analyzing the moderating effect of local industrial policies considering the turnover of local officials. Columns 1–3 use *IndLoan* as the dependent variable. Variables are defined in Table A1 in S1 Appendix. The sample period is from 2007 to 2020. In parentheses are t-statistics.

***, **, * indicate statistical significance at the 1%, 5%, and 10% level, respectively.

**Table 8. Large private shareholders and industrial loans of city commercial banks (Difference between groups).**

| | (1) | (2) |
|---|---|---|
| | *IP=0* | *IP=1* |
| | *IndLoan* | *IndLoan* |
| *LPS* | 0.145* | 0.518*** |
| | (1.67) | (6.89) |
| control variables/industry/year | Yes | Yes |
| *N* | 460 | 756 |
| F | 11.75 | 21.38 |
| r2 | 0.484 | 0.461 |
| Chi2 | 6.78 | |
| P-value | 0.0092 | |

This table reports the results of difference between groups analyzing the impact of large private shareholders on banks' industrial loans. Columns 1–2 use *IndLoan* as the dependent variable. Variables are defined in Table A1 in S1 Appendix. The sample period is from 2007 to 2020. In parentheses are t-statistics.

***, **, * indicate statistical significance at the 1%, 5%, and 10% level, respectively.

commercial banks and empirically analyze the impact of large private shareholders on city commercial banks' lending to these shareholders' industry. Importantly, we test the moderating effect of local industrial policies on this relationship. The results show that an increase in large private shareholders' shareholding encourages banks to lend more to these shareholders' industries. Furthermore, local industrial policy acts as a typical moderator: when large private shareholders' industries are supported by local industrial policies, the impact of these shareholders on encouraging city commercial banks' lending is enhanced.

We provide an in-depth understanding of the industry expertise of banks' large private shareholders. Further, we enrich the theory of information asymmetry by examining banks' industry-specific lending. Although industrial policies are exogenous, we still do not properly evaluate them through traditional DID, which is our limitation. Future research can study the topic of this paper from a corporate governance perspective. For example, researchers can use shareholder checks and balances, the board system, and the characteristics of senior executives as moderating variables, and test their effect on the relationship between large private shareholders and bank lending. In terms of macro policy, we can also formulate and use a series of macro moderators, such as monetary policy, fiscal policy, and foreign trade.

This paper draws the following inspirations:

First, it is necessary to improve the corporate governance level of city commercial banks, and effectively play the role of large private shareholders in the allocation of bank loans. In order to achieve the goal of optimizing the allocation of commercial banks' credit resources, in addition to using non-market-oriented means such as administrative intervention or policy guidance. It can also be achieved by introducing high-quality private enterprises related to economic restructuring and transformation and upgrading to hold the share of banks. This allows banks to voluntarily increase support for structural adjustment, transformation and upgrading of the real economy. Finally, the purpose of adjusting the credit structure of city commercial banks and rationally allocating financial resources is realized.

Second, it is crucial to formulate industrial policies more rationally. Industrial policies have an important influence on mobilizing the allocation of local financial resources. From another perspective, when the industries of large private shareholders are supported by industrial policies, banks can effectively cooperate with the implementation of industrial policies. Therefore, when formulating industrial policies, government departments should fully understand the industry attributes and shareholding of large private shareholders of local financial institutions. It can ensure that the industrial policies formulated are more reasonable and improve the realizability of industrial policies.

## Supporting information

**S1 Data.**
(XLSX)

**S1 Appendix.**
(PDF)

## Author Contributions

**Conceptualization:** Jie Liu.

**Data curation:** Jie Liu.

**Formal analysis:** Jie Liu, Qiaoyun Zhang.

**Methodology:** Jie Liu, Qiaoyun Zhang.

**Software:** Jie Liu.

**Writing – original draft:** Jie Liu, Qiaoyun Zhang, Chao Xiao.

**Writing – review & editing:** Jie Liu.

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
