## [Decision Letter · Decision Letter 0]

12 Aug 2022

PONE-D-22-17876Private Large Shareholders, Industrial Policy and Industrial Loans of City Commercial BanksPLOS ONE

Dear Dr. Liu,

Thank you for submitting your manuscript to PLOS ONE. After careful consideration, we feel that it has merit but does not fully meet PLOS ONE’s publication criteria as it currently stands. Therefore, we invite you to submit a revised version of the manuscript that addresses the points raised during the review process.

We look forward to receiving your revised manuscript.

Kind regards,

LeThanh Ha

Academic Editor

PLOS ONE

Journal Requirements:

Reviewers' comments:

Reviewer's Responses to Questions

**Comments to the Author**

1. Is the manuscript technically sound, and do the data support the conclusions?

Reviewer #1: Yes

2. Has the statistical analysis been performed appropriately and rigorously? 

Reviewer #1: Yes

3. Have the authors made all data underlying the findings in their manuscript fully available?

Reviewer #1: Yes

4. Is the manuscript presented in an intelligible fashion and written in standard English?

Reviewer #1: No

5. Review Comments to the Author

Reviewer #1: This paper perhaps a good paper and has contribution for body of knowledge. However, it is too lengthy. At first, I thought perhaps there is a series of robustness check to ensure the results are reliable and vigor. However, it seems not ready for publication and it has no focus. It is confusing and the write-up is too mechanical. But the topic is interesting. So, I gave a major revision as the verdict.

First of all, kindly note that introduction section plays important role to attract the attention. When the introduction is bad, most likely the reviewers will reject the paper. Unfortunately, the introduction is the weakest part of this paper.

As earlier mentioned, it is confusing. These are my suggestions:

Rather than starting with discussion about china, focus first on what kind of contribution this paper offers to the body of knowledge. Move the first paragraph of introduction to section 2 (literature review) as a new sub-section. Name it as 2.1 Ownership Expropriation and Industrial Loan in Chinese City Commercial Banks.

As I believe the selling point of the paper is the moderating variable, it is important to show the mixed findings between large shareholders and industry loan. Therefore, paragraph 2 of Introduction can be the first paragraph. However, discuss first the mixed findings between large shareholders and industry loan. The author can break it into two paragraphs, where first paragraph discusses the positive relationship, and second paragraph discusses about negative relationship.

Following that paragraph, the author can offer industry policy as the moderating variable. Elaborate also which theory supports the argument that industry policy can strengthen the relationship. It is your second or third paragraph.

Then, show why Chinese city commercial banks are interesting research setting. Why the case of Chinese city commercial banks is important for body of knowledge? How it suits the research context? How researchers from other countries can learn from the case of Chinese city commercial banks? Yes, the author can elaborate it with the existing paragraph 3.

Then follow with the contributions, which is already in the paper (paragraph 4 to 7).

Meanwhile, section 2 (the literature review) can be divided into three section. Section 2.1 is Ownership Expropriation and Industrial Loan in Chinese City Commercial Banks. Section 2.2 is Theoretical analysis (only discuss the theoretical framework that support the research framework). Section 2.3 is for Hypothesis development. Don’t mix theoretical analysis and hypothesis development.

I still don’t get which theory that used and tested. It is unclear. This is why it is important to have Section 2.2

The Data and Methodology section is also unclear. Keep it simple and straightforward. 3.1 is for Data, 3.2 Model Specification, and 3.3 is Measures. The 3.2 can provide only the final model for brevity reason.

Add 1 table for variable definition. You can add it in 3.3 or in Appendix. Then, summarize the variable definition in Page 6 t0 7. Keep it simple.

The results are exhausting and confusing. It obfuscates the main idea of the research. The authors mix and scramble all the tables. My suggestions:

Provide only the main results for hypothesis testing, which are: Table 2 Column 3 and Table 13. Just combine the Table 2 Column 3 and Table 13 as one table. Called it as 4.2 Results

Start with 4.1 descriptive statistics (Table 1) and add correlation matrix.

Add one section and called it 4.3 Robustness check. Then provide Table 4, 16, 17, 18, and 19.

Move Table 3 to section 2.1

Move Table 6 to Section 3.1 or Appendix

Delete the rest of tables.

Add one analysis for robustness check, which is Difference-in-Difference (but this is optional as your results already massive).

Add the moderation plot based on Table 13. Check Dawson (2014).

Add contribution for body of knowledge in Conclusion section.

Add limitation and suggestion for future research in conclusion section

For sure, the manuscript is too confusing.

Reference

Dawson, J. F. (2014). Moderation in management research: What, why, when, and how. Journal of business and psychology, 29(1), 1-19.

6. PLOS authors have the option to publish the peer review history of their article (what does this mean?). If published, this will include your full peer review and any attached files.

Reviewer #1: **Yes: **Rayenda Khresna Brahmana

---

## [Author Response · Author response to Decision Letter 0]

7 Sep 2022

PONE-D-22-17876

Dear Editor LeThanh Ha,

Thank you very much for giving us the opportunity to revise the manuscript we submitted to PLOS ONE. We have revised the initial version of the paper in accordance with the requirements of the journal and reviewer's comments carefully.

Your active help and the reviewer's comments are very insightful, so that the quality of our manuscript has been significantly improved.

As requested, we have submitted the Rebuttal letter (The filename is Response to Reviewers), Marked up copy of our manuscript (The filename is Revised Manuscript with Track Changes) and Unmarked version of our manuscript (Filename is Manuscript). 

Thank you so much for your help and that of reviewers. We look forward to receiving your positive feedback.

Kind regards,

Jie Liu [Corresponding author]

School of Finance, Southwestern University of Finance and Economics

liuj@smail.swufe.edu.cn

Response to reviewer #1 comments

Dear Reviewer Rayenda Khresna Brahmana,

We really appreciate you for your carefulness and conscientiousness. Your suggestions are really valuable and helpful for revising and improving our paper. According to your suggestions, we have made the following revisions on this manuscript: 

1. First of all, kindly note that introduction section plays important role to attract the attention. When the introduction is bad, most likely the reviewers will reject the paper. Unfortunately, the introduction is the weakest part of this paper.

As earlier mentioned, it is confusing. These are my suggestions:

Rather than starting with discussion about china, focus first on what kind of contribution this paper offers to the body of knowledge. Move the first paragraph of introduction to section 2 (literature review) as a new sub-section. Name it as 2.1 Ownership Expropriation and Industrial Loan in Chinese City Commercial Banks.

As I believe the selling point of the paper is the moderating variable, it is important to show the mixed findings between large shareholders and industry loan. Therefore, paragraph 2 of Introduction can be the first paragraph. However, discuss first the mixed findings between large shareholders and industry loan. The author can break it into two paragraphs, where first paragraph discusses the positive relationship, and second paragraph discusses about negative relationship.

Following that paragraph, the author can offer industry policy as the moderating variable. Elaborate also which theory supports the argument that industry policy can strengthen the relationship. It is your second or third paragraph.

Then, show why Chinese city commercial banks are interesting research setting. Why the case of Chinese city commercial banks is important for body of knowledge? How it suits the research context? How researchers from other countries can learn from the case of Chinese city commercial banks? Yes, the author can elaborate it with the existing paragraph 3.

Then follow with the contributions, which is already in the paper (paragraph 4 to 7).

Response 1：Thank you very much for your advice. I have rewritten the Introduction based on your suggestions. 

We focus first on what kind of contribution this paper offers to the body of knowledge (paragraph 1).

Indeed, the selling point of the paper is the moderating variable.Therefore, we focus on the moderating effects of industrial policy (paragraph 2). We also elaborate which theory supports the argument that industry policy can strengthen the relationship.

Then, we show why Chinese city commercial banks are interesting research setting (paragraph 3).

2. Meanwhile, section 2 (the literature review) can be divided into three section. Section 2.1 is Ownership Expropriation and Industrial Loan in Chinese City Commercial Banks. Section 2.2 is Theoretical analysis (only discuss the theoretical framework that support the research framework). Section 2.3 is for Hypothesis development. Don’t mix theoretical analysis and hypothesis development.

I still don’t get which theory that used and tested. It is unclear. This is why it is important to have Section 2.2

Response 2：Thank you for your constructive comments. We have revised the section 2 according to your comments.

Section 2.1 is Ownership Expropriation and Industrial Loan in Chinese City Commercial Banks.

Section 2.2 is Theoretical analysis. We formulate the theory of information asymmetry. This specifically includes adverse selection and moral hazard.

Section 2.3 is for Hypothesis development. 

3. The Data and Methodology section is also unclear. Keep it simple and straightforward. 3.1 is for Data, 3.2 Model Specification, and 3.3 is Measures. The 3.2 can provide only the final model for brevity reason.

Add 1 table for variable definition. You can add it in 3.3 or in Appendix. Then, summarize the variable definition in Page 6 t0 7. Keep it simple.

Response 3：We have simplified the Data and Methodology section. And added Appendix TableA1 for variable definition.

4. The results are exhausting and confusing. It obfuscates the main idea of the research. The authors mix and scramble all the tables. My suggestions:

Provide only the main results for hypothesis testing, which are: Table 2 Column 3 and Table 13. Just combine the Table 2 Column 3 and Table 13 as one table. Called it as 4.2 Results

Start with 4.1 descriptive statistics (Table 1) and add correlation matrix.

Add one section and called it 4.3 Robustness check. Then provide Table 4, 16, 17, 18, and 19.

Move Table 3 to section 2.1

Move Table 6 to Section 3.1 or Appendix

Delete the rest of tables.

Add one analysis for robustness check, which is Difference-in-Difference (but this is optional as your results already massive).

Add the moderation plot based on Table 13. Check Dawson (2014).

Response 4：Thank you for your positive comments.

We added the correlation matrix as shown in Table 3 (The Descriptive statistics section).

We provide only the main results for hypothesis testing as shown in Table 4 in 4.2 Results. And add the moderation plot (Fig.3) based on Table 4. Check Dawson (2014).

We moved Table 3 and Table 6 of the original manuscript to section 2.1 and Appendix TableA2, respectively.

In 4.3 Robustness check, we have retained Tables 16, 17, 18, and 19 from the original manuscript, which are now Tables 5, 6, 7, and 8, respectively. We did not retain Table 4 in the original manuscript, which reports the effect of China's banking regulatory policy on the impact of private large shareholders, because it is not relevant to the moderating effect of industrial policy in the Robustness test. This makes the article clearer as you emphasized.

We deleted the rest of tables of the original manuscript.

For DID, this method is often used for policy evaluation. It requires uninterrupted implementation of the policy. But the industrial policies of Chinese local governments do not meet the conditions of DID. Industrial policy is not continuous. For example, an industry that is supported in the current “Five-Year Plan” may not be supported in the next “Five-Year Plan”. However, we group the sample according to the criteria of whether the private large shareholders are supported by industrial policies. And test the impact of private large shareholders on banks' industrial loans in two groups. These results as shown in Table 9. This makes our robustness tests more richer and reliable.

5. Add contribution for body of knowledge in Conclusion section.

Add limitation and suggestion for future research in conclusion section.

Response 5：Thank you very much for your valuable comments. We have added contribution for body of knowledge and suggestions for future research in conclusion section. In addition, we have removed the inspirations in Conclusion section (the last two paragraphs of Conclusion section in the original manuscript).

6. For sure, the manuscript is too confusing.

Response 6：We removed irrelevant content and tables and rewrote the main content based on your suggestion. In addition, we have the help of language editing to ensure that our manuscripts are presented in an intelligible fashion and written in standard English.

---

## [Decision Letter · Decision Letter 1]

20 Sep 2022

PONE-D-22-17876R1Private Large Shareholders, Industrial Policies and Industrial Loans of City Commercial Banks: Evidence from ChinaPLOS ONE

Dear Dr. Liu,

Thank you for submitting your manuscript to PLOS ONE. After careful consideration, we feel that it has merit but does not fully meet PLOS ONE’s publication criteria as it currently stands. Therefore, we invite you to submit a revised version of the manuscript that addresses the points raised during the review process.

We look forward to receiving your revised manuscript.

Kind regards,

LeThanh Ha

Academic Editor

PLOS ONE

Additional Editor Comments:

The current version is too long by conducting many robustness tests but unintentionally.

Please review and make the paper more concise.

Authors also should highlight more their contributions and theoretical framework.

Reviewers' comments:

Reviewer's Responses to Questions

**Comments to the Author**

1. If the authors have adequately addressed your comments raised in a previous round of review and you feel that this manuscript is now acceptable for publication, you may indicate that here to bypass the “Comments to the Author” section, enter your conflict of interest statement in the “Confidential to Editor” section, and submit your "Accept" recommendation.

Reviewer #1: (No Response)

2. Is the manuscript technically sound, and do the data support the conclusions?

Reviewer #1: Yes

3. Has the statistical analysis been performed appropriately and rigorously? 

Reviewer #1: Yes

4. Have the authors made all data underlying the findings in their manuscript fully available?

Reviewer #1: No

5. Is the manuscript presented in an intelligible fashion and written in standard English?

Reviewer #1: No

6. Review Comments to the Author

Reviewer #1: Dear authors

First of all, thank you for all efforts made by your side. I really appreciate your revision. However, I still have minor things that you need to revise.

I still do not satisfy with the introduction, specifically, the first paragraph. As per earlier comments: it is important to show the mixed findings between large shareholders and industry loan. One good example is the paper from Graffin et al (2020). Look at their first and second paragraph. Benchmark the way they write the mixed findings. It is not necessarily to have theoretical gap like Graffin et al (2020), you always can show the empirical gap about the mixed findings between the X (independent variable) and Y (dependent variable).

Related to hypothesis development, you need to have three hypotheses. Refer to Brambor et al (2006), Balli & Sorensen (2013) and Dawson (2014) for the important to have three hypotheses in the moderating effect study. The first hypothesis is about the relationship between X and Y. The second hypothesis is about the relationship between M and Y. The last hypothesis is about the interaction term. It is up to your research group, whether want to have separate sub-section for second and third hypothesis.

The theoretical statement for moderating effect is either:

(1)“The positive relationship between X and Y would be strengthened by the moderating effect of M”

(2) “The moderating effect of M would strengthen the positive relationship between X and Y”

Or, (3) you can do several modifications like what Graffin et al (2020).

If you do not want to do DiD approach, you can argue it in the manuscript, and put it as limitation of study in the conclusion.

I thought that your earlier version has implication for policymakers and industry, but why I can read it again in this revised version? Add it.

I know that PLOSONE will make the final copy-editing, but check again the structure and grammatical errors. For example, “private large shareholders “, “This significantly differs from” (line 41), “differing from” (Line 57), and many more. To be honest, content-wise, your research is okay. However, it is still confusing and takes time to read and understand it. I leave this matter to the associate editor, how they will take action towards this issue.

Further, I know it might seem minor, but I do hope the authors really showing effort for the revision and not taking it for granted.

References

Brambor, T., Clark, W. R., & Golder, M. (2006). Understanding interaction models: Improving empirical analyses. Political analysis, 14(1), 63-82.

Balli, H. O., & Sørensen, B. E. (2013). Interaction effects in econometrics. Empirical Economics, 45(1), 583-603.

Dawson, J. F. (2014). Moderation in management research: What, why, when, and how. Journal of business and psychology, 29(1), 1-19.

Graffin, S. D., Hubbard, T. D., Christensen, D. M., & Lee, E. Y. (2020). The influence of CEO risk tolerance on initial pay packages. Strategic Management Journal, 41(4), 788-811.

---

## [Author Response · Author response to Decision Letter 1]

10 Oct 2022

Response to Academic Editor 

Dear Editor LeThanh Ha, 

We really appreciate you for your carefulness and conscientiousness. Your suggestions are really valuable and helpful for revising and improving our paper. According to your suggestions, we have made the following revisions on this manuscript: 

1.The current version is too long by conducting many robustness tests but unintentionally. 

Please review and make the paper more concise. 

Authors also should highlight more their contributions and theoretical framework.

Response 1： We edited the article again to make it more concise. For example, control variables are no longer included in the table of regression results (except Table 4). We have also deleted Table 8 in the original manuscript because its role is similar to Table 7. We have added a paragraph in the introduction (Lines 86-90), which describes more our contributions and enrichment of information asymmetry theory. 

Response to reviewer #1 

Dear Reviewer Rayenda Khresna Brahmana, 

Thank you again for your recognition of our article. Your suggestion is very helpful to improve the quality of our paper. According to your suggestion, we have made the following modifications to this manuscript: 

1.I still do not satisfy with the introduction, specifically, the first paragraph. As per earlier comments: it is important to show the mixed findings between large shareholders and industry loan. One good example is the paper from Graffin et al (2020). Look at their first and second paragraph. Benchmark the way they write the mixed findings. It is not necessarily to have theoretical gap like Graffin et al (2020), you always can show the empirical gap about the mixed findings between the X (independent variable) and Y (dependent variable). 

Response 1： Your comments are very professional and important. At present, there are not many scholars studying the relationship between large shareholders and industrial loans of banks. This leads us to be unable to effectively summarize the mixed relationship. Fortunately, the relationship between large shareholders and the total scale of bank loans has received some attention. Therefore, we show the mixed findings between them. 

2.Related to hypothesis development, you need to have three hypotheses. Refer to Brambor et al (2006), Balli & Sorensen (2013) and Dawson (2014) for the important to have three hypotheses in the moderating effect study. The first hypothesis is about the relationship between X and Y. The second hypothesis is about the relationship between M and Y. The last hypothesis is about the interaction term. It is up to your research group, whether want to have separate sub-section for second and third hypothesis. 

The theoretical statement for moderating effect is either: 

(1)“The positive relationship between X and Y would be strengthened by the moderating effect of M” 

(2) “The moderating effect of M would strengthen the positive relationship between X and Y” 

Or, (3) you can do several modifications like what Graffin et al (2020). 

Response 2： We now have three hypotheses in the manuscript based on your suggestion. 

3.If you do not want to do DID approach, you can argue it in the manuscript, and put it as limitation of study in the conclusion. 

Response 3： We explained the reason for not using DID in the manuscript, and took it as our limitation in the conclusion. 

4. I thought that your earlier version has implication for policymakers and industry, but why I can read it again in this revised version? Add it. 

Response 4： We added the inspiration of the paper, which highlights implication for policymakers and industry. 

5.I know that PLOSONE will make the final copy-editing, but check again the structure and grammatical errors. For example, “private large shareholders”, “This significantly differs from” (line 41), “differing from” (Line 57), and many more. 

Response 5： We have dealt with grammar errors again by consulting professionals and our school's English teachers.

---

## [Editor Report · Decision Letter 2]

22 Nov 2022

Large Private Shareholders, Industrial Policies and Industrial Loans of City Commercial Banks: Evidence from China

PONE-D-22-17876R2

Dear Dr. Jie Liu

We’re pleased to inform you that your manuscript has been judged scientifically suitable for publication and will be formally accepted for publication once it meets all outstanding technical requirements.

Kind regards,

LeThanh Ha

Academic Editor

PLOS ONE
---

## [Editor Report · Acceptance letter]

28 Nov 2022

PONE-D-22-17876R2 

Large Private Shareholders, Industrial Policies and Industrial Loans of City Commercial Banks: Evidence from China 

Dear Dr. Liu:

I'm pleased to inform you that your manuscript has been deemed suitable for publication in PLOS ONE. Congratulations! Your manuscript is now with our production department. 

Kind regards, 

on behalf of

Dr. LeThanh Ha 

Academic Editor

PLOS ONE